# Anion-Sensing Properties of Cyclopentaphenylalanine

**DOI:** 10.3390/molecules27123918

**Published:** 2022-06-18

**Authors:** Ivan Petters, Matija Modrušan, Nikolina Vidović, Ivo Crnolatac, Nikola Cindro, Ivo Piantanida, Giovanna Speranza, Gordan Horvat, Vladislav Tomišić

**Affiliations:** 1Department of Chemistry, Faculty of Science, University of Zagreb, Horvatovac 102a, 10000 Zagreb, Croatia; ipetters@chem.pmf.hr (I.P.); mmodrusan@chem.pmf.hr (M.M.); ncindro@chem.pmf.hr (N.C.); 2Dipartimento di Chimica, Università degli Studi di Milano, Via C. Golgi, 19, 20133 Milan, Italy; nikolina@iptpo.hr (N.V.); giovanna.speranza@unimi.it (G.S.); 3Institute of Agriculture and Tourism, Department of Agriculture and Nutrition, Karla Huguesa 8, 52440 Poreč, Croatia; 4Ruđer Bošković Institute, Bijenička 54, 10000 Zagreb, Croatia; ivo.crnolatac@irb.hr (I.C.); ivo.piantanida@irb.hr (I.P.)

**Keywords:** cyclopeptide, anion binding, fluorimetry, circular dichroism, UV spectrometry, ^1^H NMR, MD simulations, solvent effect

## Abstract

Cyclic pentaphenylalanine was studied as an efficient anion sensor for halides, thiocyanate and oxoanions in acetonitrile and methanol. Stability constants of the corresponding complexes were determined by means of fluorimetric, spectrophotometric, ^1^H NMR, and microcalorimetric titrations. A detailed structural overview of receptor–anion complexes was obtained by classical molecular dynamics (MD) simulations. The results of ^1^H NMR and MD studies indicated that the bound anions were coordinated by the amide groups of cyclopeptide, as expected. Circular dichroism (CD) titrations were also carried out in acetonitrile. To the best of our knowledge, this is the first example of the detection of anion binding by cyclopeptide using CD spectroscopy. The CD spectra were calculated from the structures obtained by MD simulations and were qualitatively in agreement with the experimental data. The stoichiometry of almost all complexes was 1:1 (receptor:anion), except for dihydrogen phosphate where the binding of dihydrogen phosphate dimer was observed in acetonitrile. The affinity of the cyclopeptide receptor was correlated with the structure of anion coordination sphere, as well as with the solvation properties of the examined solvents.

## 1. Introduction

Anion sensing by synthetic receptors is a highly dynamic ongoing topic in supramolecular chemistry due to the major impact of anion binding and transport on physical and biological processes. To date, numerous anion receptors have been synthesized and characterized, namely molecules with hydrogen donating groups [1,2,3,4], aromatic substituents or permanent positive charge [5,6,7,8,9], functionalized macrocycles such as cyclopeptides [10,11,12,13,14], calixarenes, calixpyrroles, and cryptands [15,16], or cation complexes with organic ligands [7,8,9,13,17]. These receptors differ in the affinities and selectivities for anionic species, as well as in the analytical signal by which the anion binding is detected. In the context of anion detection and transport, receptors made from biomolecules are biocompatible candidates and most probably less toxic than synthetic organic receptors. In that line, cyclopeptides are promising candidates for efficient anion sensors and anion transporters in vivo due to the abundance and orientation of anion-binding groups, including amide protons, and resistance to biodegradation when compared to their linear analogues. One of the most sensitive methods used for the detection of anion–receptor complex formation in solution is fluorimetry [15,18,19,20,21,22]. The method is fast, reliable, and gives a multivariate response that is well suited for stability constants determination. The three most fluorescent natural amino acids are tryptophan, tyrosine, and phenylalanine. We have prepared a cyclic homopentapeptide with L–phenylalanine subunits (Figure 1) to obtain a sensitive fluorescent anion sensor in solution. The fluorescence of phenylalanine is weak in comparison to that of tyrosine and tryptophan, but this can be partly prevailed by the design of the receptor containing several phenylalanine subunits. Additionally, the chosen number of subunits is in line with our previous work in which cyclopentaleucine was studied as a versatile anion binder in solution [10]. In the present work, the affinity of cyclopentaphenylalanine (**L**) towards halogen anions (Cl^−^, Br^−^, and I^−^), thiocyanate and several oxoanions (NO_3_^−^, HSO_4_^−^, and H_2_PO_4_^−^) was tested in acetonitrile and methanol by means of fluorimetric, ^1^H NMR and spectrophotometric titrations, and microcalorimetry. We observed strong changes in fluorescence of phenylalanine subunits upon anion binding, from which the stability constants of the corresponding complexes were determined. The anion coordination was investigated by molecular dynamics simulations, and the binding of almost all amide protons was observed. This was complemented by circular dichroism titrations performed in acetonitrile in which the strong change in 220 nm region was visible upon anion complexation. CD spectra were calculated from the structures obtained by MD simulations and were compared to the experimental ones. The anion–binding affinity of **L** was compared to that of similar cyclopeptides. The results are discussed with a special reference on the solvation effect of reaction participants.

## 2. Experimental Section

### 2.1. Materials

Cyclic pentaphenylalanine **L** was synthesized according to the procedure described elsewhere [23]. Acetonitrile (MeCN, Sigma-Aldrich, St. Louis, MO, USA, ≥99.9% grade) and distilled methanol (MeOH, J.T. Baker, Phillipsburg, NJ, USA, HPLC grade) were used as solvents. The salts used were tetraethylammonium chloride (TEACl, Sigma-Aldrich, >98%), tetrabuthylammonium bromide (TBABr, Sigma-Aldrich, >99%), tetrabuthylammonium iodide (TBAI, Sigma-Aldrich, >99%), tetrabuthylammonium thiocyanate (TBASCN, Sigma-Aldrich, 98%), tetrabuthylammonium nitrate (TBANO_3_, Sigma-Aldrich, 98%), tetrabuthylammonium hidrogen sulfate (TBAHSO_4_, Sigma-Aldrich, >99%), and tetrabuthylammonium dihydrogen phosphate (TBAH_2_PO_4_, Sigma-Aldrich, >99%). For the preparation of solutions used in NMR titrations, deuterated acetonitrile-d_3_ (Eurisotop, Saint-Aubin, France, 99.8%), and deuterated methanol (Eurisotop, 99.8%) were used.

### 2.2. Physicochemical Measurements

#### 2.2.1. Fluorimetric, Spectrophotometric, and Circular Dichroism Titrations

Fluorimetric titrations were conducted on Varian Cary Eclipse Fluorimeter, UV-Vis spectra were recorded by Agilent Cary 5000 spectrophotometer, and CD spectra were recorded on JASCO J815 spectrophotometer. Fluorimeter and UV-Vis spectrophotometer were connected to thermostating device, whereas CD titrations were performed at room temperature. The excitation wavelength used in all fluorimetric titrations was 260 nm, spectra were recorded at 2 nm interval with 0.4 s integration time, 10 nm excitation, and emission slits were used. In titrations with iodide, the obtained spectra were corrected by the anion concentration dependent solvent emission spectra. The sampling interval in UV-Vis experiments was 1 nm and the signal at each wavelength was averaged for 0.2 s. CD spectra were recorded in the 200–300 nm wavelength range with a 0.2 nm data pitch and scanning speed of 200 nm/min; 2 scans were accumulated for each spectrum. In all titrations, quartz cell with a 1 cm optical path length was used, and the initial volume of titrant solution was in the range from 2 to 2.3 mL. Each spectrum was collected after the stepwise addition of anion salt solution to the reaction mixture. The obtained fluorimetric and UV-Vis absorption data were processed by a multivariate regression analysis implemented in the HypSpec 1.1.50 software (Protonic Software, Leeds, United Kingdom) [24]. In the determination of stability constants of dihydrogen phosphate complexes in acetonitrile, anion dimerization was taken into the account [25]. All fluorimetric titrations were performed in triplicate.

#### 2.2.2. ^1^H NMR Titrations

^1^H NMR titrations were carried out at 25 °C by means of a Bruker Avance III HD 400 MHz with a solvent signal used as a standard. In all titrations the solution of anion salt and receptor **L** was added to the solution of **L** contained in the NMR tube. The concentration of **L** in titrant and titrand solutions was equal. Spectra were recorded at 16 pulses. ^1^H NMR spectra were processed and visualized by means of MestReNova 6.1 software (Mestrelab Research, Santiago de Compostela, Spain). Stability constants of the corresponding complexes were determined from the obtained ^1^H NMR spectra by using multivariate analysis implemented in HypNMR 4.0 software (Protonic Software, Leeds, UK) included in the Hyperquad program package [24].

#### 2.2.3. Isothermal Titration Calorimetry

Microcalorimetric measurements were conducted with an isothermal titration calorimeter Microcal VP-ITC at 25.0 °C. Thermograms were processed using the Microcal OriginPro 7.0 program (OriginLab Corporation, Northampton, MA, USA).

In the calorimetric studies of chloride complexation by **L**, the enthalpy changes were recorded upon stepwise additions of salt solution into the solution of cyclopeptide ligand (*V*_0_ = 1.4 cm^3^). The heats measured in the titration experiments were corrected for the heats of titrant dilution obtained by blank experiments. The dependence of successive enthalpy changes on the titrant volume was processed by non-linear least-squares fitting procedure using OriginPro 7.5 program (OriginLab Corporation, Northampton, MA, USA). Measurements were repeated three times.

#### 2.2.4. Solubility Determinations

Saturated solutions of **L** in acetonitrile and methanol were prepared by adding an excess amount of the solid substance to the solvent. The obtained mixtures were left in a thermostat at 25 °C for several days in order to equilibrate. The concentrations of saturated solutions were determined at 25.0 °C spectrophotometrically by means of a Varian Cary 5000 spectrophotometer equipped with a thermostatting device. Calibration curves were obtained by measuring the absorbances of solutions of known concentrations.

### 2.3. Molecular Dynamics Simulations

The MD simulations were produced using the GROMACS package (version 2019.5, University of Groningen; Stockholm Royal Institute of Technology; Uppsala University) [26,27,28,29,30,31,32]. Intramolecular and nonbonded intermolecular interactions were modelled by the Optimized Parameters for Liquid Simulations-All Atoms (OPLS-AA) force field [33]. Parameters for the hydrogen sulfate and dihydrogen phosphate anions were assigned by LigParGen server [34,35,36]. The cyclopeptide **L** and its anion complexes were solvated in cubical boxes containing about 3000 acetonitrile or 4000 methanol molecules, with periodic boundary conditions. The solvent boxes were equilibrated prior to inclusion of solute molecules with the box density after equilibration in all cases being close to the experimental one within 2%. In the simulations where anion–ligand complexes were investigated, tetramethylammonium (TMA) cation was included to neutralize the box. This counterion was kept fixed at the box periphery, whereas the initial position of the complex was at the box center. In all simulations an energy minimization procedure was performed followed by a MD simulation in *NpT* conditions for 50.5 ns, where the first 0.5 ns were not used in data analysis. The Verlet algorithm [37] with a time step of 1 fs was employed. The cut-off radius for nonbonded van der Waals and short-range Coulomb interactions was 16 Å. Long-range Coulomb interactions were treated by the Ewald method as implemented in the Particle Mesh Ewald (PME) procedure [38]. The simulation temperature was kept at 298 K with the Nosé–Hoover algorithm [39,40] using a time constant of 1 ps. The pressure was kept at 1 bar by Martyna–Tuckerman–Tobias–Klein [41] algorithm and the time constant of 1 ps. Average molecular structures of cyclopeptide–anion complexes were obtained by Principle Component Analysis (PCA) on coordination matrix whose rows contained distances and angles between heavy atoms of anions, or carbonyl oxygen atoms, in the case of free **L**, and amide nitrogen and hydrogen atoms of peptide ligands. The chosen structures were closest to the centroids of the most populous clusters in space defined by the first two or three principal components. Figures of molecular structures were created using VMD 1.9.3 software (University of Illinois, Champaign, IL, USA) [42].

## 3. Results and Discussion

### 3.1. Anion Complexes of **L** in Acetonitrile

The results of fluorimetric, spectrophotometric, ^1^H NMR, and microcalorimetric titrations, summarized in Table 1, show that the receptor **L** readily binds halogen and several polyatomic anions in acetonitrile.

The ability of cyclopeptide **L** to act as a fluorimetric anion probe was tested by fluorimetric titrations. The addition of all anions to the **L** solution was accompanied by the change in the fluorescence signal (Figure 1 and Appendix A). The binding of chloride, dihydrogen phosphate, and hydrogen sulfate anions resulted in the increase in the fluorescence signal in 280 nm region, while the binding of bromide, iodide, thiocyanate, and nitrate anions reduced fluorophore emission in this part of spectrum (Figure 2). From the results of these experiments, we managed to extract stability constants for almost all anion complexes of **L** (Table 1), except in the case of Cl^−^ anion, where the relative fluorescence intensity increased almost linearly up to the equimolar ratio. However, we were able to determine the **L**-Cl^−^ stability constant by microcalorimetry (Figure 3). In the case of **L**-I^−^ complex the presence of iodide anion caused strong quenching of **L** emission (Appendix A) which also prevented the reliable data processing. This effect is similar to the quenching of cyanophenylalanine fluorescence in the presence of iodide anion described in the work of Pazos et al. [43] The affinity of **L** towards iodide anion was determined by ^1^H NMR titrations described below.

Spectrophotometric titrations of **L** with the anion solutions were also carried out, whereby the change in the phenyl group absorption was monitored (Figure 4 and Appendix A). The changes in this region of the spectrum upon anion binding were quite modest, and in titrations with iodide, nitrate, and thiocyanate were completely covered by the absorption of the anion which resulted in the almost linear dependence of absorbance on the anion concentration (Appendix A, respectively). Additionally, at the end of these titrations a slight precipitation was observed, especially when higher concentrations of **L** were used.

The binding of anions by **L** was also monitored by ^1^H NMR spectroscopy (Figure 5 and Appendix A). In the corresponding titrations, the shift of the amide and Cα protons was observed, which was a clear indication that the anions were bound to the backbone amide groups of **L**. From these results, we derived the stability constants of iodide, thiocyanate, and nitrate complexes which are in agreement with those obtained by fluorimetry (Table 1). In the course of the titration of **L** with dihydrogen phosphate anion, amide protons were shifted downfield which was followed by the emergence of new signals of these protons at higher anion:receptor molar ratio (Appendix A). Analogous phenomenon was observed in the UV spectra where a significant change in the spectra was seen after the equimolar ratio (Appendix A). These findings could be attributed to the binding of the second dihydrogen phosphate anion to the **L**-H_2_PO_4_^−^ complex. We could not confirm the existence of 1:2 complex by the regression analysis due to the fact that the quality of obtained data was below the limit needed for reliable stability constant determination. Still, the fact that a similar complex with cyclopentaleucine and dihydrogen phosphate dimer in acetonitrile was observed in our previous investigations [10], and that dihydrogen phosphate spontaneously forms dimers in acetonitrile solution [25] corroborate our assumption.

The values of stability constants of anion complexes of cyclopentaleucine [10] and receptor **L** are similar, indicating that there is no different influence of the aliphatic or aromatic sidechains on anion binding capabilities of cyclopentapeptides. The reaction enthalpy (−15 kJ mol^−1^) and entropy (65 J mol^−1^ K^−1^) for **L**-Cl complex formation are close to the corresponding values for complexation of chloride with cyclopentaleucine (∆_r_*H* = −11 kJ mol^−1^, ∆_r_*S* = 76 J mol^−1^ K^−1^). Therefore, the inclusion of fluorescent groups to the cyclopeptide sidechain improved the anion detection capability (sensitivity) of **L** but did not alter the binding affinity compared to the cyclopentaleucine, the aliphatic sidechain analogue. The only notable difference is that receptor **L** did not form sandwich complexes with hydrogen sulfate anion, as was the case with cyclopentaleucine [10].

The structures of ligand **L** and its complexes were thoroughly investigated by molecular dynamics simulations. In the course of MD simulations of free **L**, the existence of intramolecular hydrogen bonds was observed (Figure 6a). The average number of intramolecular hydrogen bonds was 1.3, a single bond was present in the 44% of simulation time, two bonds during 31%, and three during 7% of the simulation. We performed the MD simulations of **L**–anion complexes of 1:1 stoichiometry with two starting forms, one named *endo*, in which the anions were situated above the center of cyclopeptide backbone on the same side as sidechains, and the other named *exo*, in which the anion was located on the other side of the cyclopeptide ring. In the simulations of *endo* complexes, all anions were coordinated by almost all amide protons in a pentagonal pyramidal orientation (Table 2, Figure 6b–h). On the other hand, in the simulations of *exo* complexes, the anions were coordinated by only a few amide groups and in some cases the dissociation of complex occurred. These results clearly suggest that the preferred anion binding mode was that in the *endo* form. The oxoanions were bound by one or two oxygen atoms and thiocyanate anion was exclusively bound by its nitrogen atom. The position of phenyl rings in free **L** and its complexes was also investigated by examining the distribution of distance between center of mass of phenol rings and geometric center defined by Cα atoms (Appendix A). This was taken as a measure of proximity of the aromatic substituents to the anion binding site. The results suggest that the phenolic rings are closer to anion binding site in polyatomic anion complexes compared to the free **L** whereas, in the halide complexes, aromatic subunits are kept at larger distances.

Circular dichroism titrations revealed structural changes due to chloride, bromide, dihydrogen phosphate, and hydrogen sulfate anions complexation manifested as a change in the CD spectra in the *n*-π* transition band of peptide groups (Figure 7 and Appendix A). In all cases, the strong negative signal centered at 230 nm emerged as the anion solution was added to the solution of **L**. These titrations were only carried out for the above-mentioned anions and only in acetonitrile. It is most probable that the binding of other anions would cause similar changes in CD spectra. These spectra were also calculated by DichroCalc webserver [44,45] with default settings from the structures of **L** obtained by MD simulations. In this process, every 50th structure of the **L**–anion complex was extracted from the MD trajectory, 1000 structures per simulation in total. Then, a single CD spectrum was calculated for every structure, and an average of all spectra was taken as a representative one. The experimental and calculated spectra are compared in Figure 8. A good qualitative agreement in the shape of spectra can be seen, especially in the relation between the spectrum of free **L** and its complexes. However, it should be noted that the calculated peaks are slightly blue-shifted relative to the experimental ones and differ in intensity, whereas the experimentally determined molar ellipticity coefficients are similar for all four complexes. This can be because the CD spectra were solely calculated from the structure of **L**, without taking into account the bound anion and solvent molecules, using parameters which were derived for aqueous solutions [46]. Still, this approach yielded a satisfactory agreement between the experimental and calculated data and, in a way, validated the structural MD simulations results, which demonstrated the spatial ordering of peptide groups upon anion complexation.

### 3.2. Anion Complexes of **L** in Methanol

The ability of cyclopeptide **L** to bind anions was also investigated in methanol, a protic solvent which resembles water in its hydrogen-bonding capabilities.

The stability constants of all 1:1 complexes observed in acetonitrile were determined in methanol as well by means of all three spectroscopic methods (Table 1). No binding of dihydrogen phosphate dimer with **L** was observed in methanol, although that was noticed in acetonitrile. All ions, except dihydrogen phosphate and hydrogen sulfate, formed complexes of relative low stability with log *K* being about 1.5. The formation of all complexes was detected by fluorimetric titrations. The results of these titrations were successfully used for the stability constants determinations (Table 1, Figure 9 and Appendix A). Almost all anions, except dihydrogen phosphate and hydrogen sulfate, caused quenching of the emission of **L** (peak centered at 285 nm) upon binding.

In addition, we performed ^1^H NMR titrations in methanol. From the results of these experiments, we were able to extract the stability constants of all anion complexes studied. Again, the results of ^1^H NMR titrations are in satisfactory agreement with those determined by fluorimetric titrations (Table 1, Appendix A).

Unlike in acetonitrile, the significant UV spectral changes were observed during titrations. By processing the data of these titrations, we calculated the stability constants of **L** complexes (Table 1, Figure 10 and Appendix A). In some cases, the changes in the absorption spectra were near detection limit, rendering the stability constant determination difficult. Nevertheless, the satisfactory agreement between the stability constants determined by this method and those obtained by fluorimetric and ^1^H NMR titrations corroborate the obtained results.

Overall, the affinity of receptor **L** towards anions is weaker in methanol than in acetonitrile. The decrease in stability constants of complexes in methanol compared to acetonitrile is about 4.5 in the log *K* value for chloride, 3.5 for bromide, and between 1 and 2 for other anions. In order to thermodynamically characterize these differences, we constructed thermodynamic cycles for the complexation reactions in acetonitrile and methanol connected by the transfer of reactants and products from one solvent to the other (Figure 2 and Appendix A). The standard reaction Gibbs energies of complexation reactions were calculated from the titrations data, the transfer Gibbs energy of **L** was determined from the ligand solubility measurements, and the anion transfer Gibbs energies were taken from the literature [46]. The transfer of the **L**–anion complex (**L**A^−^) was then easily calculated by the equation:∆_t_*G*°(**L**A^−^, MeCN→MeOH) = ∆_t_*G*°(A^−^, MeCN→MeOH) + ∆_t_*G*°(**L**, (MeCN→MeOH) + ∆_r_*G*°(MeOH) − ∆_r_*G*°(MeCN)(1)
where ∆_t_*G*° denotes standard transfer Gibbs energy and ∆_r_*G*° stands for standard reaction Gibbs energy of complexation. The experimentally determined solubilities of free **L** in acetonitrile ((2.72 ± 0.02) × 10^−3^ mol dm^−3^) and in methanol ((2.81 ± 0.03) × 10^−3^ mol dm^−3^) are very similar. Consequently, the transfer Gibbs energy of **L** is negligible. The transfer of **L**–anion complexes from acetonitrile to methanol is slightly favorable for the **L**-Cl^−^ complex. In the case of bromide, iodide, nitrate, and thiocyanate complexes, the transfer of these species is accompanied with small absolute values of ∆_t_*G*°. On the other hand, the Gibbs energies of free anion transfer from methanol to acetonitrile range from −8 to −29 kJ mol^−1^. Therefore, the main contribution to the decrease in the stability of **L**–anion complexes in methanol comes from the strong anion solvation properties of that solvent compared to acetonitrile. The observed difference in the overall complex stability is mainly a consequence of the free anion solvation which is significantly stronger in methanol than in acetonitrile.

Molecular dynamics simulations in methanol yielded similar results compared to those in acetonitrile. Again, *endo* complexes of anions are preferred, whereby the anions are bound by almost all amide protons (Table 3) in a pentameric pyramidal coordination sphere (Appendix A). The change in solvent had an effect on the number of intramolecular hydrogen bonds. In methanol, 0.97 bonds were observed on average, which is about 25% lower than the number of bonds observed in acetonitrile. This is expected and in line with better hydrogen-acceptor and donor properties of methanol compared to acetonitrile.

## 4. Conclusions

The anion binding ability of pentaphenylalanine cyclopeptide **L** in acetonitrile and methanol was thoroughly characterized by experimental and computational methods. The results showed that this compound could act as a sensitive anion sensor due to its fluorescence properties, whereby the affinity for anion binding was moderate to strong in the organic solvents used. In acetonitrile, compound **L** exhibited the strongest affinity for chloride anion, followed by that for bromide, hydrogen sulfate, and dihydrogen phosphate. Stability constants of **L**–halogen anion complexes are strongly correlated with the anion size, whereby the affinity of **L** towards larger anions is weaker (Table 1). Receptor **L** almost exclusively forms 1:1 complexes in this solvent. The only exception pertains to the complex with dihydrogen phosphate, where the formation of 1:2 species is also likely to take place (indications of this process were observed by ^1^H and NMR titrations). When methanol was used as a solvent, a moderate to high loss of affinity and selectivity of **L** was observed, mostly due to the stronger solvation of free anions in MeOH compared to MeCN. The reason for the loss of anion selectivity in methanol can be deduced from the thermodynamic cycles presented in this work. It is evident that the differences in the standard Gibbs complexation energies of **L** with anions in acetonitrile are almost the same as the differences in the transfer Gibbs energies for the same anions. These effects cancel out upon transfer of all reaction participants from acetonitrile to methanol rendering the complexation Gibbs energies for halogen, nitrate, and thiocyanate close in value.

The binding of anions induces the conformational changes of cyclopeptide receptor, which can be readily detected by circular dichroism spectroscopy. The CD spectra were calculated from the MD structures of **L** complexes, and the main spectral characteristics were similar to those found experimentally. These findings corroborate the molecular-level description of conformational changes observed in the MD simulations. Furthermore, the results of these simulations suggest that the amide groups are bound to the anion by their hydrogen atoms, and the coordination sphere of the anion is pentagonal pyramidal.

The presented results suggest that the combination of the anion-binding ability of cyclopentapeptide structural motif and fluorescence properties of phenylalanine results in a receptor which is a versatile anion binder in protic and aprotic solvents, and it is consequently a possible candidate for anion sensing and transport.

## Data Availability

The data presented in this study are available in Appendix A.

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
