# Peer review of "Anion-Sensing Properties of Cyclopentaphenylalanine"

_molecules, 2022, doi:10.3390/molecules27123918_

Round 1

Reviewer 1 Report

The authors have conducted a compelling study of anion-sensing properties of a cyclopeptide. Through a combined approach using spectroscopic, calorimetric, and computational techniques, they provided a comprehensive and detailed view of molecular basis of anion selectivity of cyclopentaphenylalanine. In particular, their analysis using thermodynamic cycles furnishes interesting molecular insights into the origin of the peptide’s selectivity for anions and its connection to solvation free energy of anions in solvent. Overall, this study could fill in some gap of our understanding of anion-cyclopeptide interactions. Therefore, I recommend this paper to be considered for publication in Molecules as long as my following minor comments could be addressed. No further review is needed.

1)    The data presented in Table 1 is of the most importance. In particular, they suggest that the affinity of the cyclopeptide appears sensitive to the size difference between halide ions and insensitive to structural difference from other polyatomic anions in MeCN but the other way around in MeOH. Can the authors comment on this point and its molecular origin? The thermodynamic cycle analysis has revealed some interesting results and the authors have attributed this trend mainly to different solvation free energy of bare anions in different solvents. Yet, I also noticed some other trends such as the one that DG_r is almost the same for different halide anions in MeOH but vastly different in MeCN. How these trends could be associated with different sizes of halide ions or with different structures of mono-/polyatomic anions would be highly interesting. The authors may consider perform analysis on their MD simulations to examine the interactions between solvent molecules and bare/bound anions to seek possible hints.

2)    Cyclopolypenylalanine is known to be prone to self-assemble into nanostructures. Can the authors comment on how its self-assembly property influences its anion-sensing ability and how the presence of anions would impact its self-assembly?

3)    On p. 8, the author mentioned that a peptide conformation (shown in Fig. 6a) is used as the model conformation for studying complex formation. Is this conformation, which has all its sidechains on the same side of backbone, the only major conformation of this peptide? Is this true for different solvent environments?

Author Response

1)    The data presented in Table 1 is of the most importance. In particular, they suggest that the affinity of the cyclopeptide appears sensitive to the size difference between halide ions and insensitive to structural difference from other polyatomic anions in MeCN but the other way around in MeOH. Can the authors comment on this point and its molecular origin? The thermodynamic cycle analysis has revealed some interesting results and the authors have attributed this trend mainly to different solvation free energy of bare anions in different solvents. Yet, I also noticed some other trends such as the one that DG_r is almost the same for different halide anions in MeOH but vastly different in MeCN. How these trends could be associated with different sizes of halide ions or with different structures of mono-/polyatomic anions would be highly interesting. The authors may consider perform analysis on their MD simulations to examine the interactions between solvent molecules and bare/bound anions to seek possible hints.

The reason for the loss of selectivity of L towards halogen, nitrate and thiocyanate anions in methanol with respect to acetonitrile lies in the solvation properties of methanol. The difference in the anion solvation in acetonitrile and methanol is directly reflected in the values of the anion transfer Gibbs energies.

We have included the following paragraph to the conclusion:

The reason for the loss of anion selectivity in methanol can be deduced from the thermodynamic cycles presented in this work. It is evident that the differences in the standard Gibbs complexation energies of L with anions in acetonitrile are almost the same as the differences in the transfer Gibbs energies for the same anions. These effects cancel out upon transfer of all reaction participants from acetonitrile to methanol rendering the complexation Gibbs energies for halogen, nitrate and thiocyanate anions close in value.

As for the dihydrogen phosphate and hydrogen sulfate anions, the decrease of the affinity of L towards these species in methanol cannot be discussed on the basis of free anion solvation, we didn't found the values of the transfer Gibbs energies for these species in literature. The molecular origin of the stronger anion solvation in methanol compared to the acetonitrile surely lies in the ability of methanol to form hydrogen bonds with free anions, which is not a property of acetonitrile. We agree that the analysis of free and bounded anion solvation in MD simulations could lead to some hints on the observed differences in the anion affinities in two solvents used. In order to fully computationally describe anion solvation, we would need to calculate the effects of the cavity formation in both solvents and interactions between solvent molecules in anion solvation spheres. To obtain a reliable estimate of all these contributions by computation is not a straightforward procedure, we would like to interpret the obtained results with respect to the experimentally determined transfer Gibbs energies of reaction participants.

2)    Cyclopolypenylalanine is known to be prone to self-assemble into nanostructures. Can the authors comment on how its self-assembly property influences its anion-sensing ability and how the presence of anions would impact its self-assembly?

We tried, but could not find the reference to the Cyclopolypenylalanine in the literature, what we found is the report on the synthesis of linear poly(phenylalanine).(10.1021/acsami.1c13013) We have some preliminary results on the complexation properties of linear peptides, usually linear peptides tend to bind anions with lower affinity compared to their cyclic analogues. One possible mechanism for the affinity enchantment of poly(phenylalanine) towards anions could be the assembly of polypeptide in nanostructures in which the amide groups are close enough to form a binding site similar to the ones in cyclic peptides. As for the anion impact on self-assembly of polypeptide, it is known that the binding of the chloride anion to linear peptides promotes the formation of the quasi-cyclic structure, that effect was studied in detail in our previous work.( 10.1021/acs.orglett.0c00036) The similar effect of the structure ordering can surely be expected upon anion binding to poly(phenylalanine).

3)    On p. 8, the author mentioned that a peptide conformation (shown in Fig. 6a) is used as the model conformation for studying complex formation. Is this conformation, which has all its sidechains on the same side of backbone, the only major conformation of this peptide? Is this true for different solvent environments?

In the MD simulations of free cyclopentaphenylalanine in acetonitrile and methanol the dominant structures we observed had sidechains mostly oriented to the same side of the cyclopeptide ring. It seems that the ring made of five amino acids does not have the freedom to allow the conformations with one or more Calpha atoms located on the other side of the cyclopeptide ring, probably due to intramolecular hydrogen bonding.

Reviewer 2 Report

Petters et al. presented a comprehensive host-guest study between cyclopentaphenylalanine and different kinds of anions. The binding constant study is detailed. Especially, they observed signal change in circular dichroism, which may be associated with the conformation change to cyclopentaphenylalanine when host-guest complexes were formed.   

My main concern is that anion-sensing properties will be significantly influenced by electrostatic interaction. In this case, the authors did not characterize the pH of cyclopentaphenylalanine solution. But I believe the pH will be critical for its binding constant with anions. For example, if the pH is low enough to protonate, the formed protonated cationic species will no doubt have stronger interactions with anions. That may also influence the current conclusion that cyclopentaphenylalanine can form a 1:2 complex with dihydrogen phosphate. I suggest the author determine the pH of all the studied solutions, which will be important to verify the correctness of their binding constant.

Author Response

My main concern is that anion-sensing properties will be significantly influenced by electrostatic interaction. In this case, the authors did not characterize the pH of cyclopentaphenylalanine solution. But I believe the pH will be critical for its binding constant with anions. For example, if the pH is low enough to protonate, the formed protonated cationic species will no doubt have stronger interactions with anions. That may also influence the current conclusion that cyclopentaphenylalanine can form a 1:2 complex with dihydrogen phosphate. I suggest the author determine the pH of all the studied solutions, which will be important to verify the correctness of their binding constant.

The only protonable sites of the cyclopentaphenylalanine are the backbone secondary amide groups. The secondary amides are very weak Bronsted bases (N-Methylbenzamide has a pKa value of -1.7, 10.1139/v65-069), which means that the protonation occurs at very high concentrations of H+ ions, half ionisation occurs in 34 % sulphuric acid. We doubt that the H+ concentrations in neat acetonitrile (pKautoprotolysis ≈ 27) and methanol (pKautoprotolysis = 16.5) are sufficient to promote secondary amide group protonation. (10.1351/pac198759121693)

Reviewer 3 Report

Dear Editor,

 I have read the manuscript entitled: “Anion-sensing properties of cyclopentaphenylalanine” and I would like to address following suggestions to the authors:

In the manuscript the references should be formatted conform to the requirements of the Molecules journal ([ ]).

 At Line 190, what the authors mean by “S7-S12”?.  Where are the figures S1-S4 (missing)?

Minor points:

I ask the authors to check spelling and others grammatical errors:

Line 22, with the exception of = except for; Line 92, volume of titrand = volume of titrant; Line 175, all anion to =all anions to;

Author Response

In the manuscript the references should be formatted conform to the requirements of the Molecules journal ([ ]).

We have formatted the references as instructed.

 At Line 190, what the authors mean by “S7-S12”?.  Where are the figures S1-S4 (missing)?

 The figures S1-S4, and also S5 and S6 are related to the fluorimetric titrations and are called in the paragraph above in the line 176.

Minor points:

I ask the authors to check spelling and others grammatical errors:

Line 22, with the exception of = except for; Line 92, volume of titrand = volume of titrant; Line 175, all anion to =all anions to;

We have included suggested corrections in the manuscript.